# Analyses of Virulence Genes of *Clavibacter michiganensis* subsp. *michiganensis* Strains Reveal Heterogeneity and Deletions That Correlate with Pathogenicity

**DOI:** 10.3390/microorganisms9071530

**Published:** 2021-07-18

**Authors:** Miryam Valenzuela, Marianela González, Alexis Velásquez, Fernando Dorta, Iván Montenegro, Ximena Besoain, Francisco Salvà-Serra, Daniel Jaén-Luchoro, Edward R. B. Moore, Michael Seeger

**Affiliations:** 1Laboratorio de Microbiología Molecular y Biotecnología Ambiental, Departamento de Química, Universidad Técnica Federico Santa María, Valparaíso 2390123, Chile; alexvelasquezsaez@gmail.com; 2Centro de Biotecnología “Dr. Daniel Alkalay Lowitt”, Universidad Técnica Federico Santa María, Valparaíso 2390136, Chile; fernando.dorta@usm.cl; 3Instituto de Química, Pontificia Universidad Católica de Valparaíso, Valparaíso 2373223, Chile; gonzalezveg4n@gmail.com; 4Escuela de Obstetricia y Puericultura, Facultad de Medicina, Universidad de Valparaíso, Valparaíso 2540064, Chile; ivan.montenegro@uv.cl; 5Escuela de Agronomía, Pontificia Universidad Católica de Valparaíso, Quillota 2260000, Chile; ximena.besoain@pucv.cl; 6Department of Infectious Diseases, Institute for Biomedicine, Sahlgrenska Academy, University of Gothenburg, SE-41346 Gothenburg, Sweden; francisco.salva.serra@gu.se (F.S.-S.); daniel.jaen.luchoro@gu.se (D.J.-L.); erbmoore@ccug.se (E.R.B.M.); 7Culture Collection University of Gothenburg (CCUG), Sahlgrenska Academy, University of Gothenburg, SE-41346 Gothenburg, Sweden; 8Microbiology, Department of Biology, University of the Balearic Islands, 071 22 Palma de Mallorca, Spain

**Keywords:** *Clavibacter michiganensis* subsp. *michiganensis*, tomato bacterial canker, virulence assay, virulence genes, cellulase, strain diversity, genome deterioration

## Abstract

*Clavibacter michiganensis* subsp. *michiganensis* (*Cmm*) is the causal agent of bacterial canker of tomato. Differences in virulence between *Cmm* strains have been reported. The aim of this study was the characterization of nine *Cmm* strains isolated in Chile to reveal the causes of their differences in virulence. The virulence assays in tomato seedlings revealed different levels of severity associated with the strains, with two highly virulent strains and one causing only mild symptoms. The two most virulent showed increased cellulase activity, and no cellulase activity was observed in the strain causing mild symptoms. In three strains, including the two most virulent strains, PCR amplification of the 10 virulence genes analyzed was observed. In the strain causing mild symptoms, no amplification was observed for five genes, including *celA*. Sequence and cluster analyses of six virulence genes grouped the strains, as has been previously reported, except for gene *pelA1*. Gene sequence analysis from the genomes of five Chilean strains revealed the presence of deletions in the virulence genes, *celB, xysA*, *pat-1*, and *phpA*. The results of this study allow us to establish correlations between the differences observed in disease severity and the presence/absence of genes and deletions not previously reported.

## 1. Introduction

Bacterial canker of tomato caused by *Clavibacter michiganensis* subsp. *michiganensis* (*Cmm*) is the most important bacterial disease affecting tomato crops in Chile, due to severe symptoms and its easy dissemination. Disease symptoms include wilting, chlorosis, stem canker, vascular necrosis, and brown spotting surrounded by creamy-white halos on the fruit. However, differences in the expression of symptoms have been observed in the field as well as under controlled laboratory conditions [1,2].

Several genes have been associated with the virulence of *Cmm* [3,4] such as genes that encode serine-proteases (*chpC*, *pat-1*, *phpA*, and *phpB*), carbohydrate-active enzymes (CAZymes), including cellulases (*celA* and *celB*), xylanases (*xysA* and *xysB*), and pectinases (*pelA1* and *pelA2*), and a tomatinase, encoded by the *tomA* gene. Studies on the presence of putative virulence genes carried out in different countries have shown variations among analyzed strains [2,5,6,7,8,9]. The absence of some genes (e.g., *celA* and *pat-1*) was correlated, in some cases, with a delay in the expression of symptoms or in a reduction or total loss of virulence [10,11,12].

On the other hand, it has been demonstrated that genomes of bacterial pathogens are reduced in the process of permanent association with the host [13]. This process is associated with an increase in numbers of pseudogenes [14]. However, Gartemann et al. [3] detected a low number of pseudogenes in the strain NCPPB382, suggesting that this pathogen is in the process of adaptation to the plant host.

Previously, a characterization of 25 Chilean strains of *Cmm* was carried out, using a multilocus sequence analysis (MLSA)/multilocus sequence typing (MLST) analysis based on five housekeeping genes [15], which resulted in detection of a low degree of heterogeneity between the strains. The strains were classified into three groups, with most of the strains delineated within one of the groups. A subsequent comparative genome analysis of three Chilean strains, each strain representing one of the different MLSA/MLST groups, revealed that most virulence genes were conserved, except *pelA1*, which exhibited more variability in different strains, and the *phpA* and *phpB* genes, which were absent in one strain [16]. The aim of this study was the genomic characterization of the virulence of nine *Cmm* strains isolated in Central Chile to reveal the causes of their differences in virulence. In this study, we compared the virulence observed in tomato plants and explored the gene repertoire, sequence variability, and presence of pseudogenes of virulence genes in Chilean strains belonging to the three MLST/MLSA groups. The results of this study showed correlations between virulence, pathogenicity genes’ repertoire, and the presence of pseudogenes.

## 2. Materials and Methods

### 2.1. Bacterial Strains and Culture Conditions

*Clavibacter michiganensis* subsp. *michiganensis* (*Cmm*) strains used in this study are listed in Table 1. Strains were obtained from the Phytopathology Laboratory Culture Collection of Chile (Escuela de Agronomía, Pontificia Universidad Católica de Valparaiso, Chile) and from the culture collection of the Laboratory of Molecular Microbiology and Environmental Biotechnology (Chemistry Department, Universidad Técnica Federico Santa María, Valparaiso, Chile). These *Cmm* strains had been previously identified and characterized through microbiological, biochemical, and molecular techniques [15]. *Cmm* strains were routinely cultured in yeast-peptone-glucose agar (YPGA; 5 g L ^−1^ yeast extract, 5 g L^−1^ Bactopeptone, 10 g L^−1^ glucose, 15 g L^−1^ agar) and incubated at 28 °C for 72 h.

### 2.2. Virulence Assay in Tomato Seedlings

Nine Chilean *Cmm* strains were selected, based on previous pathogenicity tests [15], that showed different levels of disease symptoms. The virulence assay was performed as described previously [16]. Briefly, tomato seeds cv. San Pedro were germinated in pots with peat and maintained at 20–25 °C in a growth chamber with a photoperiod of 16 h for 3 weeks. *Cmm* strains were plated on YPGA and incubated for 3 days at 28 °C. Bacterial biomass was resuspended in sterile distilled water and adjusted to 10^8^ CFU mL^−1^. Bacterial concentration was confirmed, using the dilution plate method. For each *Cmm* strain, five tomato seedlings were inoculated by puncturing the stem with a sterile needle 1.0 cm above the cotyledons and adding 10 µL of bacterial suspension onto the wound. Plants inoculated with sterile distilled water were used as negative controls. Disease symptoms were monitored every week. The assay was repeated three times. At day 21, plants were classified, based on disease symptoms: 0 = no symptoms; 1 = canker in the site of inoculation; 2 = canker extended by the stem; 3 = yellowing and slight wilting; 4 = wilting in all leaves; 5 = dead plant. The disease index (DI) was calculated, using the following formula:$$\mathrm{DI}=\frac{\sum \;(\mathrm{Severity}\;\mathrm{rating}\;\times \;\mathrm{Number}\;\mathrm{of}\;\mathrm{plants}\;\mathrm{in}\;\mathrm{that}\;\mathrm{rating})}{\mathrm{Total}\;\mathrm{number}\;\mathrm{of}\;\mathrm{plants}\;\times \;\mathrm{higher}\;\mathrm{rating}}\;\times \;100$$

### 2.3. Statistical Analysis

Virulence assay data were analyzed. Normality and homogeneity of variances were first checked through Shapiro–Wilk test and Levene test, respectively. Then, they were analyzed by a one-way analysis of variance (ANOVA). The means were compared by Tukey’s HSD test (*p* ≤ 0.05) using R version 3.6.3. Uninoculated controls were excluded from statistical analysis.

### 2.4. Assay for Cellulase Activity

Endocellulase activity was evaluated, using the method described by Meletzus et al. [17], with modifications. Bacterial strains were grown overnight in Yeast-Peptone-Glucose Broth (YPGB, 5 g L^−1^ yeast extract; 5 g L^−1^ bactopeptone; 10 g L^−1^ glucose) in a rotatory shaker at 150 rpm and 30 °C. Turbidity was measured by spectrophotometer and all cultures were diluted in YPGB to adjust the turbidity to 0.3 at 600 nm. Subsequently, 10 μL of each culture were plated onto TCYA medium (10 mL L^−1^ tomato juice; 1 g L^−1^ yeast extract; 5 g L^−1^ carboxymethylcellulose; 15 g L^−1^ agar) for cellulase activity. Plates were incubated for 4 days at 28 °C. Plates of TCYA medium were stained with Congo red (1 g L^−1^) for 20 min and finally bleached with 1 M NaCl. Enzymatic activity was compared by measuring the size of the halo around bacterial growth. Three replicates for each medium were carried out and the experiment was done three times.

### 2.5. PCR-Amplification of Virulence Genes

Ten virulence genes were selected for PCR-amplification, using primer pairs for amplification of *chpC* and *tomA* [5], and *celA* and *pat-1* [2] (Table 2). For *phpA*, *phpB*, *celB*, *xysA*, *pelA1*, and *pelA2* gene amplifications, primer pairs were designed, based on the genome sequence of *Cmm* strain NCPPB382 as reference [1], using the Primer-BLAST tool [18] (Table 2).

PCR assays were carried out, using a total volume of 30 μL, containing 2 µL DNA template from bacterial lysates [15], 15 µL GOTaq Master Mix 2X (Promega, Madison, WI, USA), 1 μL of each primer (10 µM), and 11 µL of nuclease free water. The amplification program included DNA denaturation at 95 °C for 5 min, followed by 30 cycles of DNA denaturation at 95 °C for 1 min, annealing at 55–60 °C (Table 2) for 1 min, extension at 72 °C for 1 min, and a final extension step at 72 °C for 10 min. The PCR products were checked by electrophoresis in 1% agarose gel. Gels were stained, using SafeView (NBS biologicals, Cambridgeshire, UK). PCR assays were performed twice, using different lysates as templates. 

### 2.6. Sequence and Cluster Analysis of Virulence Genes

PCR products of six genes, including *celA*, *celB*, *chpC*, *pat-1*, *pelA1*, and *tomA* of the nine *Cmm* strains, were purified and sequenced by Unidad de Secuenciación, Pontificia Universidad Católica de Chile (Santiago, Chile), using the primers described (Table 2). Sequences were edited, assembled, and aligned manually, using the Vector NTI v10 software (Invitrogen, Carlsbad, CA, USA). The sequences obtained were aligned, using MUSCLE [19], and compared with reference gene sequences from strain NCPPB382. Phylogenetic trees were constructed with MEGA6 software [20], using the neighbor joining algorithm [21], with bootstrap values based on 1000 replications [22].

### 2.7. Analysis of Virulence Genes and Their Proteins in Cmm Genomes

Three available genomes’ sequences of Chilean *Cmm* strains were previously determined and analyzed [16]. Carbohydrate-Active enZymes (CAZymes), including CelA, CelB, PelA1, PelA2, XysA, and XysB, of the complete and closed genomes of Chilean *Cmm* strains MSF322 and VL527 [16] were checked and compared with reference strain NCPPB382, using the Carbohydrate-Active EnZymes database (CAZy; [23]). For Chilean strains OP3 (draft genome), VQ28 and VQ143, the presence and the analysis of CAZymes were carried out manually. The virulence gene sequences, and their respective protein sequences, were extracted from Chilean strains and reference strain including *celA*, *celB*, *pelA1*, *pelA2*, *xysA, xysB*, *pat-1*, *phpA*, and *phpB*. The nucleotide sequence identity between Chilean strains and reference strain NCPPB382 was obtained, using the NCBI-BLAST tool (blastn and BLAST genome). Sequence comparisons were performed, using MUSCLE. The domain prediction was obtained based on protein sequences, using Interpro [24], and protein representation were carried out, using DOG2.0 [25].

## 3. Results

### 3.1. Disease Severity Varied between the Strains

*Cmm* strains analyzed in this study exhibited different virulence levels on tomato seedlings cv. San Pedro (Figure 1), ranging from a canker at the site of inoculation to severe wilting observed 21 days post-inoculation. The strain with the highest disease index (DI) value was the strain VL527 with DI = 65, followed by the strain MSF322 with DI = 59. Similar DI values were obtained for the strains VQ519, VL542, and OP3 (DI = 46, 45, and 48, respectively), and for strains VQ28 and OP7 (DI = 38 and 37, respectively). The lowest DI value obtained was for the strain VQ143 (DI = 11), followed by strain VL368 (DI = 29). The results of the statistical analysis of virulence assay on tomato seedlings cv. San Pedro are shown in Figure 2.

### 3.2. Cellulase Activity Was Higher in More Virulent Strains

Cellulase activity was detected through halos observed around bacterial growth in the plate cultures for most *Cmm* strains, except for the strain VQ143 (Figure 3). The biggest halo was observed for strain VL527, followed by strains MSF322 and VL542. Similar halo sizes were observed for strains VQ28, VL368, and OP3. The smallest halos were observed for the strains VQ519 and OP7 (Table 3).

### 3.3. Virulence Genes’ Repertoire and Cluster Analysis Increased Variation between Strains

The presence of virulence genes varied among the nine *Cmm* strains analyzed (Table 3). Strains MSF322, VL527, and VL542 possessed the 10 virulence genes analyzed. Only the *celB*, *xysA*, *chpC*, and *tomA* genes were detected by PCR analysis in all nine strains. Strains VL368 and VQ519 were PCR-negative for the *phpA* and *phpB* genes, whereas strains OP3 and OP7 were PCR-negative for the *pelA2*, *phpA*, and *phpB* genes. Strain VQ28 was PCR-negative for the *pat-1*, *phpA*, and *phpB* genes, while strain VQ143 was PCR-negative for the *celA*, *pelA1*, *pelA2*, *phpA*, and *phpB* genes.

The nucleotide sequences of PCR products were analyzed for the virulence genes *celA*, *celB*, *chpC*, *pat-1*, *pelA1*, and *tomA* (GenBank accessions MZ356262 to MZ356312). The percentage of polymorphic sites among Chilean strains and NCPPB382 was low in most virulence genes. The highest value of polymorphism was obtained for *pelA1* with 3.3% (16/489), followed by *celB* (1.6%; 17/1092) and *celA* (1.1%; 10/914). Less than 1% was obtained for *tomA* (0.2%; 1/497) and *chpC* (0.2%; 1/584). No polymorphism was observed in the *pat-1* genes of the eight strains. 

Cluster analysis for five virulence genes was carried out (Figure 4), to visualize groupings among strains. The *pat-1* gene was not included in this analysis due to the absence of polymorphism. Each gene showed some differences in strain grouping. The *celA, celB*, and *tomA* gene analysis clustered Chilean strains in two, three, and one group, respectively, separated from strain NCPPB382. The *chpC* and *pelA1* gene analysis clustered Chilean strains in two and four groups, respectively, one group including strain NCPPB382. 

### 3.4. Deletions Detected in the Sequences of Virulence Genes celB, xysA, pat-1, and phpA Truncate the Respective Proteins

CAZy.org database was consulted to obtain information about strains NCPPB382, VL527, and MSF322. Cellulase CelA is present in the strains NCPPB382, MSF322, and VL527, although CelB is present only in strains NCPPB382 and VL527. Based on PCR analysis and BLAST, the presence of *celA* and *celB* was confirmed in strains VL527, MSF322, OP3, and VQ28. In strain VQ143, the *celA* gene was absent, but *celB* was detected. The analysis of complete sequences of *celB* obtained from genome data allowed the detection of a deletion. The *celB* sequences of strains VL527 and NCPPB382 had 3G in positions 315–317, and in strains MSF322, OP3, VQ2,8 and VQ143 one G was missing (GGG→GG). This deletion caused a frameshift, and a stop codon truncated the protein after the aa 134, in the endoglucanase domain (Figure 5). The same deletion was found in the genome sequences of 15 other strains available in the NCBI Database (IDs: QLMX01000066.1, QLMU01000024.1, MDHK01000065.1, MDHM01000001.1, MDHC01000001.1, MDHD01000277.1, QLMV01000029.1, QLMZ01000063.1, RDQW01000006.1, QLNC01000027.1, QLMM01000006.1, QLMO01000004.1, QLMQ01000008.1, MZMP01000001.1, and CP033724.1). The presence of xylanases XysA and XysB was confirmed in the strains NCPPB382 and MSF322 in the CAZy.org database. Nevertheless, the presence of XysB alone was confirmed in strain VL527. Based on PCR analysis and BLAST, the presence of *xysA* and *xysB* was confirmed in strains VL527, MSF322, OP3, VQ28, and VQ143. The analysis of *xysA* gene sequences revealed a deletion. The *xysA* sequences of strains MSF322 and NCPPB382 had 4G in positions 1241–1244, whereas in strains VL527, OP3, VQ28, and VQ143 one G was missing (GGGG→GGG). This deletion changed the frameshift, and a stop codon truncated the protein after the aa 418, almost at the end of the xylanase domain (Figure 5). The same deletion was found in the sequences of 11 other strains available in the NCBI Database (IDs: MDHK01000023.1, MDHE01000015.1, MDHF01000313.1, MDHL01000001.1, QLMV01000001.1, QLMY01000005.1, QLML01000002.1, QLNB01000001.1, QLMN01000001.1, QLMQ01000001.1, and QLMR01000001.1). The presence of pectinases PelA1 and PelA2 was confirmed in strains NCPPB382, VL527, and MSF322 in the CAZy.org database. The presence of the *pelA1* and *pelA2* genes was confirmed in strains VL527, MSF322, OP3, and VQ28, but not found in strain VQ143. In order to confirm the results obtained by PCR analysis for the absence of virulence genes, a search in the available genome sequences of Chilean strains was carried out for the *pelA2*, *phpA*, and *phpB* genes, which were not detected by PCR in strain OP3, and the *pat-1*, *phpA*, and *phpB* genes, which were not detected by PCR in strain VQ28. BLAST analysis showed 98.4%, 97.7%, and 98.2% identity (100% sequence coverage in all cases) for *pelA2* in strains MSF322, VL527, and VQ28, respectively, and 93.2% (97% sequence coverage) in strain OP3. For *pat-1*, the percentage of nucleotide identity was 100% (100% sequence coverage) in strains VL527 and OP3, and 99.8% identity (100% sequence coverage) in strain MSF322; no significant match was found for the gene in strain VQ28. The analysis of the gene sequences of *pat-1* revealed two deletions. The sequence of *pat-1* had 7G in positions 693–699 and 4T in positions 803–806 in all strains analyzed, except for the strain MSF322 that lost a G and a T, respectively (GGGGGGG→GGGGGG; TTTT→TTT). The first deletion caused a frameshift, and a stop codon truncated the protein after the aa 239, in the peptidase domain (Figure 5). This deletion was not found in the sequences of other strains in the NCBI Database. For *phpA* the nucleotide identity was 99.5%, and 100% (both 100% sequence coverage) for strains MSF322 and VL527, respectively, and no significant match was found for strains OP3 and VQ28. The analysis of gene sequences of *phpA* showed four deletions in the gene sequence of strain MSF322, at positions 551–556T (TTTTT→TTTT), 695–699T (TTTTT→TTTT), 737–741G (GGGGG→GGGG), and 794–798T (TTTTT→TTTT), where one nucleotide was missing in each case. The first deletion caused a frameshift, and a stop codon truncated the protein after the aa 189, in peptidase domain (Figure 5). For *phpB* the nucleotide identity was 100% for strains MSF322 and VL527 (both 100% sequence coverage). BLAST analysis for the *phpB* gene in the genomes indicated the absence of this gene in strains OP3 and VQ28. The identities of nucleotide sequences between Chilean *Cmm* strains and reference strain NCPPB382 and the deletions detected are shown in Table 4.

## 4. Discussion

In this study, an analysis of *Clavibacter michiganensis* subsp. *michiganensis* virulence genes of nine Chilean strains and their potential correlation with expressed symptoms and cellulase activity was carried out. 

The results of virulence assays showed differences between *Cmm* strains in the expression of plant symptoms, ranging from severe to nearly asymptomatic. Variations in virulence between strains have been reported in other studies [1,2]. Previously, we reported the results of virulence assays for the strains VL527, MSF322, and OP3 [16]. Taking all results together, the strain that showed the most severe symptoms was VL527 followed by MSF322. Intermediate symptoms were observed with the strains VQ28, VQ519, VL542, OP3, and OP7; mild symptoms with strain VL368; and almost no symptoms were observed with strain VQ143. Based on these results, the virulence of the strains was not associated with groups defined by MLSA/MLST analysis, because VQ28, VQ143, VL368, VQ519, OP3, and OP7 belong to ST32; MSF322 belongs to ST36; and VL527 and VL542 belong to ST18 (Table 1). Cellulase activity was also observed to be different among strains; the most virulent strain, VL527, demonstrated the highest cellulase activity. No cellulase activity was observed in the less virulent strain VQ143. Endocellulase activity was detected only in strains that harbored plasmid pCM1, indicating that endocellulase activity is plasmid encoded [17].

Virulence is known to be dependent on the presence of virulence genes on the chromosome and plasmids of *C. michiganensis* [3]. In order to compare the repertoire and sequence variation of virulence genes between Chilean strains, PCR amplification and sequencing of the selected main virulence genes were carried out. The results of PCR and sequencing analyses of virulence genes revealed additional diversity between the strains belonging to ST32 group, represented by six strains. The lack of *celA* in strain VQ143 explained the low virulence observed in plants and almost null cellulase activity. The *celA* gene encodes for an endocellulase, which is a pathogenicity determinant that plays a major role in virulence and is located in the plasmid pCM1 in reference strain NCPPB382. The inactivation of the *celA* gene resulted in a loss of pathogenicity [11,12]. The lack of the *pat-1* gene in strain VQ28 was not a factor in loss of virulence, because the plant symptoms observed were similar to those in plants inoculated with other *Cmm* strains carrying the *pat-1* gene. The *pat-1* gene encodes a serine protease, which plays a role in pathogenicity and is located on the plasmid pCM2 of reference strain NCPPB382. The deletion of the sequence associated with the *pat-1* gene has been shown to significantly reduce virulence but does not resulted in a complete loss of virulence [10]. The *celA* and *pat-1* genes are transcribed extensively by *Cmm* strain NCPPB382 during early stages of tomato infection [26]. The *phpA* and *phpB* genes are homologs of the *pat-1* gene and are also located in plasmid pCM2 of reference strain NCPPB382. It was shown that the *phpA* and *phpB* genes do not induce disease symptoms in tomato plants [27]. The *pelA1* and *pelA2* genes encode for pectate lyases and are located in the *chp/tomA* region of the pathogenicity island (PAI) of reference strain NCPPB382. The *pelA1* gene has an important role in pathogenicity, unlike the *pelA2* gene, which has no effect on symptom expression [2]. An increase in the transcription of the *pelA1* gene of strain NCPPB382 during tomato infection has been reported [26]. Other researchers have reported variations in the presence of virulence genes between different strains. Kleitman et al. [5] analyzed the presence of virulence genes in pathogenic and non-pathogenic strains including the *celA* and *pat-1* genes. All the strains analyzed, including five non-pathogenic strains, were PCR-positive for the *celA* gene. Only two non-pathogenic strains were PCR-negative for the *pat-1* gene, suggesting that these genes are conserved in *C. michiganensis* populations. The study of Milijasevic-Marcic et al. [6] revealed differences in virulence between Serbian strains. Four strains showed an attenuated virulence and were also negative in the PCR test for the *pat-1* gene. The authors suggest that the attenuated virulence is caused by the loss of *pat-1* and, possibly, the loss of the plasmid carrying this gene. Bella et al. [7] analyzed strains that were PCR-negative for the *pat-1* gene. These strains were able to cause disease, but in a delayed manner. No significant difference in the disease index (DI) was observed between strains with and without the *pat-1* gene. In the study of *Cmm* strains isolated in Sicily, carried out by Ialacci et al. [8], all strains were *celA* gene PCR-positive, although the *pat-1* gene was not detected in four strains. These four strains were still pathogenic, although with reduced virulence. Tancos et al. [9] determined that some strains lacked the *phpA*, *pat-1*, and *celA* genes, reporting that *phpA* is the most common gene to be lost, whereas *celA* is the most stable gene. No correlation with virulence and loss of *phpA* was observed. Interestingly, one strain lacking *celA* and *phpA* was pathogenic for tomato, while two strains that lacked *celA* showed a lower disease incidence. All pathogenic *Cmm* strains analyzed by Thapa et al. [2] possessed a plasmid similar to pCM1 (carrying the *celA* gene); plasmid cured derivatives failed to induce bacterial canker symptoms on tomato, indicating the essential role of pCM1 in pathogenicity. In the same study, the absence of the pCM2-like plasmid (carrying the *pat-1* gene) in some pathogenic strains indicated that this plasmid is not required for pathogenicity. The variable results obtained by different authors, especially in the importance of the *pat-1* gene, could be resolved through the study of the complete genome sequences of the strains analyzed, which allows us to search for genes with greater efficiency and detect variations or deletions in the gene sequences and correlate this information with disease symptoms.

Cluster analysis using virulence gene sequences showed the same strain grouping of strains obtained in previous analysis [15,16,28], except for the *pelA1* gene, wherein the strain VQ519 separated from the group of strains belonging to ST32. Some studies of variability of virulence genes have reported low polymorphism values. In the study of Milijasevic-Marcic et al. [6], no polymorphisms were found in the *chpC* gene and only two variations in the *tomA* gene. Tancos et al. [9] evaluated the variability of virulence genes *celA, tomA, chpE, pat-1*, and *phpA*. No polymorphic sites in the *chpE, pat-1*, and *phpA* genes were observed. Low variation was observed in the *celA and tomA* genes, with polymorphism levels of 1.33% and 0.786%, respectively. Interestingly, high variability has been reported for the *pat-1* gene [29]. Mendez et al. [16] carried out a genome comparative study focused on virulence genes of *Cmm*. This study revealed that most virulence genes are highly conserved, having more sequence variability in the *pat-1, celA*, and *pelA1* genes. In the same study [16], we analyzed the repertoire of virulence genes of *Clavibacter michiganensis* subsp. *chilensis*, strain CFBP8217, which is a non-pathogenic strain associated with tomato seed [30]. This strain lacks all virulence genes analyzed in this study, except *xysB*. The comparison between strain CFBP8217 and strain VQ143 showed that VQ143 also lacked most of the virulence genes studied, but the genes *pat-1*, *celB*, and *xysA* were present.

Genome analyses of strains MSF322, VL527, OP3, VQ28, and VQ143 and comparison with reference strain NCPPB382 confirmed the absence of *phpA* and *phpB* genes in strains OP3, VQ28, and VQ143. However, the *pelA2* gene was present in strain OP3, although the sequence similarity was lower compared with other genes. This lower similarity might explain why it was not amplified by PCR. The *pelA1* and *pelA2* genes were also confirmed to be absent in the strain VQ143. Interestingly, deletions were found in the sequences of genes *celB*, *xysA, pat-1*, and *phpA*, which truncated the corresponding proteins. Deletion in the nucleotide 316G of the *celB* gene was found in strains MSF322, OP3, VQ28, and VQ143, but not in strain VL527. Deletion in the nucleotide 1243G of the *xysA* gene was found in strains VL527, OP3, VQ28, and VQ143, but not in MSF322. Four deletions were found in the *phpA* gene of strain MSF322. In strain VL527, the *phpA* gene showed 100% identity with reference strain NCPPB382 and, as mentioned above, it was absent in the strains OP3, VQ28, and VQ143. The deletion in the position 316G in the *celB* gene was reported already for the type strain LMG7333^T^ and confirmed for other three *Cmm* strains by Hwang et al. [12]. We also found this deletion in the genome sequence of 15 *Cmm* strains, other than Chilean strains, available in the NCBI database. The *celB* gene encodes for an endocellulase that is located in the chromosome outside of the *chp*/*tomA* PAI [3]. The expression of the *celB* gene was up-regulated during infection of tomato with *Cmm* strain NCPPB382, and the level of transcription was diminished by the presence of the *celA* gene [26]. Hwang et al. [12] showed that *celB* might not be important in disease development and cellulase activity, because it lacked signal peptide (SP) and, therefore, it cannot be secreted. However, the analysis of domain prediction based on protein sequences of the Chilean strains showed the presence of a signal peptide domain (Figure 5). The presence of a functional CelB protein in strain VL527 could explain the observed higher cellulase activity. However, additional analyses, such as quantification of *celA* and *celB* gene expression, will be necessary to confirm this hypothesis.

To our knowledge, this is the first report about deletions in the *xysA*, *pat-1*, and *phpA* genes in *Cmm* strains. The deletion detected in the *xysA* gene was also found in 12 *Cmm* genome sequences available in the NCBI database. The *xysA* gene, along with the *xysB* gene, encoded xylanases and was located in the chromosome, outside of the *chp*/*tomA* PAI [3]. Transcription of these genes was induced at a lower level, compared with other genes, during tomato infection with strain NCPPB382 [26].

The *pat-1* gene deletion found in Chilean strain MSF322 was not observed in other available genomes in NCBI, although we observed several other deletions in different *Cmm* strains. Burger et al. [27] determined that the motif GDSGG was required for the virulent phenotype of *pat-1* and, thus, Pat-1 could be a functional protease. The deletion in the strain MSF322 was located in the codons related with this motive but causing a change from GGG to GGC, which codified for the same amino acid glycine (the last glycine of the motif). However, this deletion also resulted in a stop codon (TGA) in the position 706–708 of the *pat-1* gene sequence truncating the protein. The *phpA* gene deletions in the Chilean strain MSF322 and the absence of the *phpA* and *phpB* genes in strains OP3 and VQ28 were in accordance with the absence of these genes in most *Cmm* strains with available genome in NCBI, as has been reported by Mendez et al. [16].

The results showed that virulence genes are suffering deletions that lead to an increase of pseudogenes, even causing the loss of virulence genes in some cases. This phenomenon was observed with genes that had homologs in the genome, such as *celA/celB*, *pat-1/phpA/phpB/chpC*, and *xysA/xysB*. Gartemann et al. [3] analyzed the presence of pseudogenes in reference strain NCPPB382, reporting that the number of pseudogenes in *Cmm* NCPPB382 was low compared to other closely related pathogens, such as *C. sepedonicus* and *Leifsonia xyli* subsp. *xyli*, and speculated that *Cmm* is a “recent” pathogen that is in the process of adapting to the host plant. It is known that the genomes of bacterial pathogens experience a genome reduction during the process of permanent association with the host and, in this process, there is an increase in the number of pseudogenes [9,10,13,14,31]. 

## 5. Conclusions

In this study, the virulence of nine Chilean strains of *Clavibacter michiganensis* subsp. *michiganensis* was analyzed. Virulence assays showed differences between *Cmm* strains in the expression of plant symptoms, ranging from severe to nearly asymptomatic. A correlation was observed with endocellulase activity, where the most virulent strain demonstrated the highest cellulase activity and no cellulase activity was observed in the less virulent strain. The analysis of the repertoire and sequence variation of virulence genes among the strains revealed additional diversity among the strains that belonged to the same group determined previously by MLST analysis. The strain that was nearly asymptomatic and without cellulase activity lacked the *celA* gene, confirming the essential role of the *celA* gene in pathogenicity. On the other hand, the absence or non-functionality of the *pat-1* gene did not show to be a determining factor in the virulence of the strains. Several deletions were found in different virulence genes, most of them reported for the first time, and the complete loss of virulence genes in some cases. This phenomenon was observed with genes that had homologs in the genome. Based on our observations, we postulate that *Clavibacter michiganensis*, as a species, continues experiencing a genome reduction in its process of permanent association with the tomato plant host.

## Figures and Tables

**Figure 1 microorganisms-09-01530-f001:**
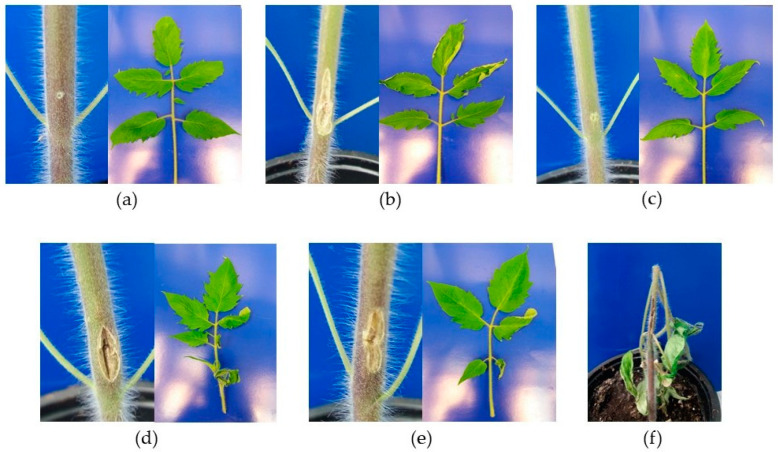
Virulence assay of Chilean *Cmm* strains in tomato seedlings cv. San Pedro. Disease symptoms of tomato seedlings 21 days after inoculation. Different levels of damage were observed in stems (canker in site of inoculation) and leaves (wilting) in (**a**) negative control and strains (**b**) VQ28, (**c**) VQ143, (**d**) MSF322, (**e**) OP3, and (**f**) VL527.

**Figure 2 microorganisms-09-01530-f002:**
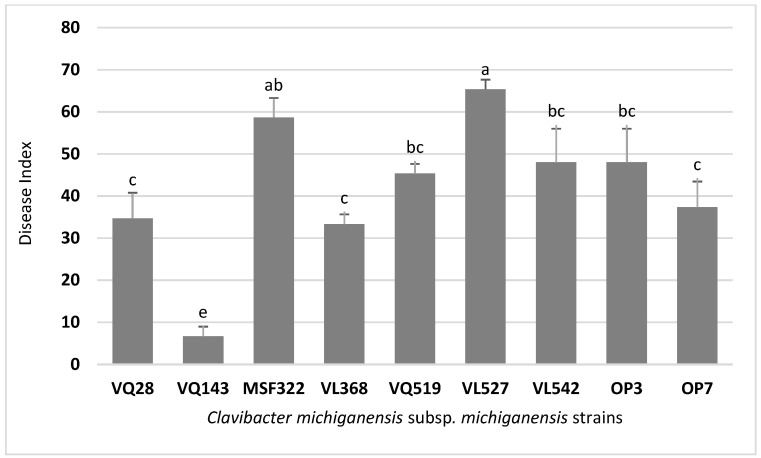
Results of virulence assay on tomato cv. San Pedro inoculated with nine Chilean *C. michiganensis* subsp. *michiganensis* strains. Data are expressed as mean ± standard deviation (*n* = 15). Means with different letters indicate significant differences from each other (*p* ≤ 0.05). Values obtained from the uninoculated control (zero) were excluded from the analyses of variance.

**Figure 3 microorganisms-09-01530-f003:**
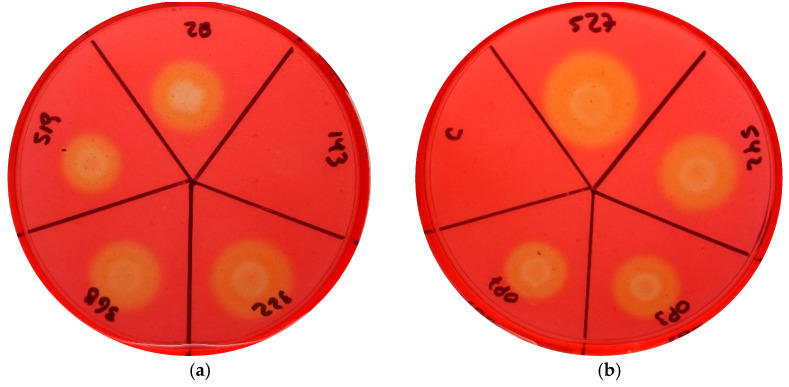
Cellulase activity of the nine *Clavibacter michiganensis* subsp. *michiganensis* strains. Bacterial isolates were grown and their concentration adjusted. After 4 days of incubation, TCYA plates were stained with Congo Red. The size of halos was measured. (**a**) Plate containing the strains (from above in clockwise order): VQ28, VQ143, MSF322, VL368, and VQ519. (**b**) Plate containing the strains (from above in clockwise order): VL527, VL542, OP3, OP7, and negative control.

**Figure 4 microorganisms-09-01530-f004:**
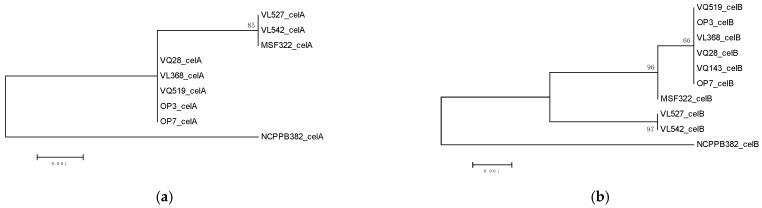
Cluster analysis of pathogenicity genes (**a**) *celA*, (**b**) *celB*, (**c**) *chpC*, (**d**) *pelA1*, and (**e**) *tomA* for nine Chilean *Clavibacter michiganensis* subsp. *michiganensis* strains and reference strain NCPPB382.

**Figure 5 microorganisms-09-01530-f005:**
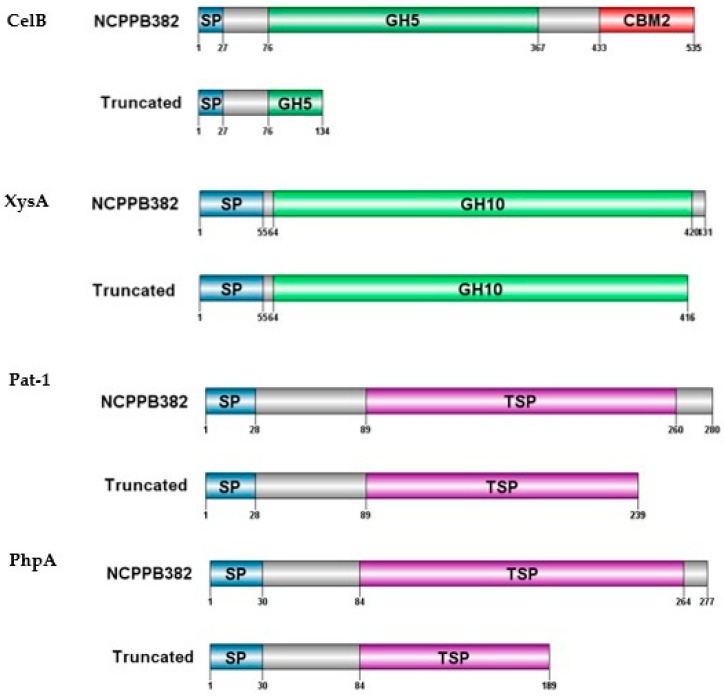
Representation of truncated proteins based on deletions detected in virulence gene sequences. SP: signal peptide; GH5: glycosyl hydrolase 5; CBM2: carbohydrate binding module; GH10: glycosyl hydrolase 10, TSP: trypsin serine-protease.

**Table 1 microorganisms-09-01530-t001:** *Clavibacter michiganensis* subsp. *michiganensis* strains used in this study and their respective Sequence-Type group determined by Multilocus Sequence Type analysis (ST-MLST) [15].

Strain	Origin (Region/Locality)	Year of Isolation	Sequence Type (ST)-MLST	GenBank Accession (Genome)
VQ28	Valparaíso/Quillota	1996	32	CP076349-CP076351
VQ143	Valparaíso/Quillota	2000	32	CP076352-CP076353
VL368	Valparaíso/Limache	2007	32	---
VQ519	Valparaíso/Quillota	2011	32	---
OP3	O’Higgins/Pichidegua	2015	32	WTCS00000000
OP7	O’Higgins/Pichidegua	2015	32	---
MSF322	Maule/Sagrada Familia	2005	36	CP047051–CP047053
VL527	Valparaíso/Limache	2012	18	CP047054–CP047055
VL542	Valparaíso/Limache	2013	18	---

--- Genome not sequenced.

**Table 2 microorganisms-09-01530-t002:** Virulence genes of *Clavibacter michiganensis* subsp. *michiganensis* and primers used in this study.

Gene	Gene Product	Gene Location	Primer Sequences (5′→3′)	Tm (°C)	Amplicon Size (bp)	Reference
*chpC*	Pat-1 type serine protease	PAI	F: GCTCTTGGGCTAATGGCCG	60	639	[5]
			R: GTCAGTTGTGGAAGATGCTG			
*tomA*	Tomatinase	PAI	F: CGAACTCGACCAGGT TCTC	60	529	[5]
			R: GGTCTCACGATCGGATCC			
*celA*	Cellulase	pCM1	F: GTAGGGCACGCATTTCAGAG	58	1240	[2]
			R: CAATGTCCTTCTTCGCCAGG			
*pat-1*	Pat-1 type serine protease	pCM2	F: TGTAGACCGTATAGCCCGTG	55	850	[2]
			R: CCTGAGACCTATTACCGCCC			
*phpA*	Pat-1 type serine protease	pCM2	F: CATTGGGTTGCTGTGTCGTT	60	605	This study
			R: GAACGTTTCCGCTTCGACTTC			
*phpB*	Pat-1 type serine protease	pCM2	F: GAGAACCAGCCTTCCCGTTC	60	596	This study
			R: CCACGAATCCTCCTGAGTCG			
*celB*	Cellulase	Chromosome	F: GGCTCGACAAGATCACCCTC	60	1283	This study
			R: ACCGACATGGACGGTCTGA			
*xysA*	Xylanase	Chromosome	F: CGATTCGACTTCTCGGGCAT	60	683	This study
			R: TCGTCCGGGTTCGAGTAGAT			
*pelA1*	Pectinase	PAI	F: AGAACGTGATCATCGGCTCG	60	520	This study
			R: TGTTCGAAGAGGATGGTGGC			
*pelA2*	Pectinase	PAI	F: ATCAACCATCTCGACCCTCCC	60	685	This study
			R: GTAACTGAAGTCGCACACCC			

**Table 3 microorganisms-09-01530-t003:** Detection of virulence genes by PCR and cellulase activity of the nine Chilean *Clavibacter michiganensis* subsp. *michiganensis* strains.

		Cell Wall-Degrading Enzymes					Serine-Proteases				Tomatinase	
Strain	Sequence Type	*celA*	*celB*	*pelA1*	*pelA2*	*xysA*	*chpC*	*pat-1*	*phpA*	*phpB*	*tomA*	Halo Diam. *
VQ28	32	+	+	+	+	+	+	-	-	-	+	17
VQ143	32	-	+	-	-	+	+	+	-	-	+	0
VL368	32	+	+	+	+	+	+	+	-	-	+	17
VQ519	32	+	+	+	+	+	+	+	-	-	+	13
OP3	32	+	+	+	-	+	+	+	-	-	+	17
OP7	32	+	+	+	-	+	+	+	-	-	+	17
MSF322	36	+	+	+	+	+	+	+	+	+	+	19
VL527	18	+	+	+	+	+	+	+	+	+	+	21
VL542	18	+	+	+	+	+	+	+	+	+	+	18

* Halo diameter average (mm) observed in the cellulase activity assay. +, detected; -, not detected

**Table 4 microorganisms-09-01530-t004:** Identity, coverage, and detection of deletions in virulence genes *celA, celB, pelA1, pelA2, xysA, xysB, pat-1, phpA, and phpB* of Chilean *Clavibacter michiganensis* subsp. *michiganensis* strains based on their genome sequences.

Strain	Gene	%ID NCPPB382	Coverage	Observations
MSF322	*celA*	99.55	100%	
	*celB*	98.88	100%	1 deletion, pseudogene
	*pelA1*	97.77	100%	
	*pelA2*	98.36	100%	
	*xysA*	99.54	100%	
	*xysB*	99.85	100%	
	*pat-1*	99.76	100%	2 deletions, pseudogene
	*phpA*	99.52	100%	4 deletions, pseudogene
	*phpB*	100.0	100%	
VL527	*celA*	99.55	100%	
	*celB*	99.07	100%	
	*pelA1*	100.0	100%	
	*pelA2*	97.65	100%	
	*xysA*	99.54	100%	1 deletion, pseudogene
	*xysB*	99.90	100%	
	*pat-1*	100.0	100%	
	*phpA*	100.0	100%	
	*phpB*	100.0	100%	
OP3	*celA*	99.64	100%	
	*celB*	98.82	100%	1 deletion, pseudogene
	*pelA1*	98.59	100%	
	*pelA2*	93.16	97%	
	*xysA*	99.54	100%	1 deletion, pseudogene
	*xysB*	99.85	100%	
	*pat-1*	100.0	100%	
	*phpA*	NF	--	
	*phpB*	NF	--	
VQ28	*celA*	99.64	100%	
	*celB*	98.82	100%	1 deletion, pseudogene
	*pelA1*	98.59	100%	
	*pelA2*	98.24	100%	
	*xysA*	99.54	100%	1 deletion, pseudogene
	*xysB*	99.85	100%	
	*pat-1*	NF	--	
	*phpA*	NF	--	
	*phpB*	NF	--	
VQ143	*celA*	NF	--	
	*celB*	98.82	100%	1 deletion, pseudogene
	*pelA1*	NF	--	
	*pelA2*	NF	--	
	*xysA*	99.54	100%	1 deletion, pseudogene
	*xysB*	99.85	100%	
	*pat-1*	100.0	100%	
	*phpA*	NF	--	
	*phpB*	NF	--	

NF: Not found; --: No coverage, because the gene was not found.

## Data Availability

The partial virulence gene sequences of the strains were deposited in GenBank, accession numbers MZ356262 to MZ356312. Genome sequences were also deposited in GenBank. Accessions by strain are listed in Table 1.

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
