# Peer review of "Analyses of Virulence Genes of Clavibacter michiganensis subsp. michiganensis Strains Reveal Heterogeneity and Deletions That Correlate with Pathogenicity"

_microorganisms, 2021, doi:10.3390/microorganisms9071530_

Round 1

Reviewer 1 Report

Authors have presented a well-structured paper and thorough description of Clavibacter michiganensis subsp. michiganensis strains in Chile and have related their findings to the Cmm strain NCPBP382.

The work is well designed and experimental details are clearly described.  A few omissions in the introduction and discussion could be addressed to provide greater benefit to authors and readers.

It is not clear why authors failed to connect their findings with previously reported MLST studies and genetic characterizations of naturally occurring weakly virulent and avirulent strains associated with tomato seed.  There is no mention of the described type strain for Clavibacter michiganensis subsp. chiloensis, a strain from Chile which represents a large number of seed-associated C. michiganensis strains isolated from tomato seed grown in Chile and elsewhere.  Avirulent and weakly virulent strains of Cm were reported on naturally infected tomato seed as early as 1994 and as new technologies became available better descriptions of such strains appeared in the literature in 2014-2015-2018.  The presence of such strains on internationally transported tomato seed is important to the seed industry.   

 Numerous avirulent/hypovirulent strains associated with tomato plants and  seed are clearly distinguishable from Clavibacter michiganensis subsp. michiganensis based on phylogenetic analyses.   Obviously it was not the intent of the present work to undertake a comparative sequence analysis of the nine key strains reported in this study with other Clavibacter subspecies or with Cmm strains from a world-wide collection and the failure to do so should in no way detract from the solid work presented in the present paper.  However, as the type strain of Clavibacter michiganensis subsp. chiloensis, is available in major culture bacterial collections, the existence of this strain should not be ignored. 

Detailed comments

A well-written, thoroughly reviewed manuscript with very few comments needed.

The authors can be commended for a detailed analysis and clear, concise description of findings.

Line: 

47-48: additional citations needed.

55: citations needed.

344: correct the grammar

449: should be:   virulence assays

Author Response

A: We would like to thank the reviewer for the careful review of our manuscript and the valuable suggestions. All the comments enclosed were attended and the manuscript was carefully checked according to the editorial suggestions.

Authors have presented a well-structured paper and thorough description of Clavibacter michiganensis subsp. michiganensis strains in Chile and have related their findings to the Cmm strain NCPBP382.

The work is well designed and experimental details are clearly described.  A few omissions in the introduction and discussion could be addressed to provide greater benefit to authors and readers.

It is not clear why authors failed to connect their findings with previously reported MLST studies and genetic characterizations of naturally occurring weakly virulent and avirulent strains associated with tomato seed.  There is no mention of the described type strain for Clavibacter michiganensis subsp. chiloensis, a strain from Chile which represents a large number of seed-associated C. michiganensis strains isolated from tomato seed grown in Chile and elsewhere.  Avirulent and weakly virulent strains of Cm were reported on naturally infected tomato seed as early as 1994 and as new technologies became available better descriptions of such strains appeared in the literature in 2014-2015-2018.  The presence of such strains on internationally transported tomato seed is important to the seed industry.   

 Numerous avirulent/hypovirulent strains associated with tomato plants and  seed are clearly distinguishable from Clavibacter michiganensis subsp. michiganensis based on phylogenetic analyses.   Obviously it was not the intent of the present work to undertake a comparative sequence analysis of the nine key strains reported in this study with other Clavibacter subspecies or with Cmm strains from a world-wide collection and the failure to do so should in no way detract from the solid work presented in the present paper.  However, as the type strain of Clavibacter michiganensis subsp. chiloensis, is available in major culture bacterial collections, the existence of this strain should not be ignored. 

A: We greatly appreciate your comments and suggestions. In a previous work about comparative genomics between different Clavibacter species and subespecies, we included Clavibacter michiganensis subsp. chilensis strain CFBP8217. The phylogenetic analyses showed that this strain is closer to C.m. subsp. phaseoli than C.m. subsp. michiganensis. Additionally, we compared the gene repertoires and we confirmed that strain CFBP8217 lacks most of virulence strains. In order to include your observation, we added the following paragraph:

“In the same study [16], we analyzed the repertoire of virulence genes of Clavibacter michiganensis subsp. chilensis, strain CFBP8217, which is a non-pathogenic strain associated with tomato seed [30]. This strain lacks all virulence genes analyzed in this study, except xysB. The comparison between strain CFBP8217 and strain VQ143, shows that VQ143 also lacks most of the virulence genes studied, but the genes pat-1, celB and xysA are present.”

Detailed comments

A well-written, thoroughly reviewed manuscript with very few comments needed.

The authors can be commended for a detailed analysis and clear, concise description of findings.

Line: 

47-48: additional citations needed.

A: Citations added

55: citations needed.

A: Citations added

344: correct the grammar

A: Grammar was improved

449: should be:   virulence assays

A: the word was corrected

Reviewer 2 Report

The main objective of this article is to establish a correlation between the repertoire of virulence genes and the pathogenicity of nine strains of Clavibacter michiganensis, a tomato pathogen, isolated in Chile. To do this, genomic and phenotypic analyses were combined with pathogenicity tests on tomato plants.  It was thus established that the repertoire of virulence genes is very variable between the different strains. These studies revealed that only the celA gene, encoding a cellulase, seems to be a determinant of pathogenicity: overall, the pathogenicity of the strains seems to be correlated with the overall cellulase activity produced by the strains and the absence of celA results in an attenuation of pathogenicity. However, no quantification of celA gene expression is presented and no genetic validation of the role of this gene in pathogenicity is presented. This could have been achieved for example by inactivating the celA gene in the most virulent strain (VL527) or transferring this gene in trans via a low copy number plasmid into the least virulent strain (VQ143) lacking cellulase activity and not containing the active celA gene. Furthermore, the results presented on the other virulence genes are speculative and not very enlightening as to their implications in pathogenicity; the variable results obtained by different authors are just mentioned without explanation.

The discussion is very long and could be considerably reduced

Author Response

A: We would like to thank the reviewer for the careful review of our manuscript and the valuable suggestions. All the comments enclosed were attended and the manuscript was carefully checked according to the editorial suggestions.

The main objective of this article is to establish a correlation between the repertoire of virulence genes and the pathogenicity of nine strains of Clavibacter michiganensis, a tomato pathogen, isolated in Chile. To do this, genomic and phenotypic analyses were combined with pathogenicity tests on tomato plants.  It was thus established that the repertoire of virulence genes is very variable between the different strains. These studies revealed that only the celA gene, encoding a cellulase, seems to be a determinant of pathogenicity: overall, the pathogenicity of the strains seems to be correlated with the overall cellulase activity produced by the strains and the absence of celA results in an attenuation of pathogenicity. However, no quantification of celA gene expression is presented and no genetic validation of the role of this gene in pathogenicity is presented. This could have been achieved for example by inactivating the celA gene in the most virulent strain (VL527) or transferring this gene in trans via a low copy number plasmid into the least virulent strain (VQ143) lacking cellulase activity and not containing the active celA gene. Furthermore, the results presented on the other virulence genes are speculative and not very enlightening as to their implications in pathogenicity; the variable results obtained by different authors are just mentioned without explanation.

 A: we appreciate your comments and suggestions. The objective of this work, as you very well describe it, is the study, at genetic and genomic level, of the repertoire of virulence genes of 9 Chilean strains of Cmm to establish correlations with the pathogenicity observed in tomato plants. Our study provides new evidence about the importance of each of the virulence genes reported, which ones would be determinant for the pathogenicity of the strains, and which could have a role in the severity of the symptoms. In addition, it accounts for the variability present in some genes and the deterioration suffered by some of them. The quantification of celA gene (and celB gene), indeed will be very helpful to stablish if the symptoms expression and the cellulase activity observed in strain VL527 it is due to a higher expression of the gene or to the presence of both functional genes. This observation was included in the text of the manuscript. The inactivation of celA gene, and the role of each of the CelA and CelB proteins domains was addressed by Hwang et al (2019). The inactivation and transfer of genes in Clavibacter strains is a very complex process, that we hope to implement it in the future. The variable results obtained by different authors, could be explained, at least in part, because most of them used PCR technique to detect the genes, and this could lead to error when the gene presents some variations or deletions in its sequence. This possible explanation was added to the text. Although much remains to be studied, we consider that our work is a contribution to the understanding of the role of virulence genes in Cmm.

The discussion is very long and could be considerably reduced

A: The discussion was reduced.

Round 2

Reviewer 2 Report

TRhe modifications incorporated into the manuscript provide answers to my essential questions